# Deepening Physical Exercise Intervention Protocols for Older People with Sarcopenia Following Establishment of the EWGSOP2 Consensus: A Systematic Review

**DOI:** 10.3390/geriatrics10040091

**Published:** 2025-07-04

**Authors:** Eduard Minobes-Molina, Sandra Rierola-Fochs, Carles Parés-Martínez, Pau Farrés-Godayol, Mirari Ochandorena-Acha, Eva Heras, Jan Missé, Fabricio Zambom-Ferraresi, Fabiola Zambom-Ferraresi, Joan Ars, Marc Terradas-Monllor, Anna Escribà-Salvans

**Affiliations:** 1Research Group on Methodology, Methods, Models and Outcomes of Health and Social Sciences (M3O), Faculty of Health Sciences and Welfare, Center for Health and Social Care Research (CESS), University of Vic-Central University of Catalonia (UVIC-UCC), 08500 Vic, Spain; eduard.minobes@uvic.cat (E.M.-M.); pau.farres@uvic.cat (P.F.-G.);; 2Institute for Research and Innovation in Life Sciences and Health in Central Catalonia (IRIS-CC), 08500 Vic, Spain; 3Spanish Society of Geriatrics and Gerontology, 28006 Madrid, Spain; 4Servei Envelliment i Salut Servei Andorrà d’Atenció Sanitària, AD500 Andorra la Vella, Andorra; eheras@saas.ad (E.H.);; 5Navarrabiomed, Hospital Universitario de Navarra (HUN), Universidad Pública de Navarra (UPNA), IdiSNA, 31008 Pamplona, Spain; fabricio.zambom.ferraresi@navarra.es (F.Z.-F.);; 6CIBER of Frailty and Healthy Aging (CIBERFES), 28029 Madrid, Spain; 7RE-FiT Barcelona Research Group, Vall d’Hebron Institute of Research (VHIR), Parc Sanitari Pere Virgili, 08023 Barcelona, Spain; 8Aging Research Center, Department of Neurobiology, Care Sciences and Society (NVS), Karolinska Institutet, Stockholm University, 171 77 Stockholm, Sweden

**Keywords:** sarcopenia, exercise, older adults, EWGSOP2

## Abstract

Background/Objectives: Sarcopenia is an age-related muscle disease that reduces strength and function in older adults. Exercise is a key intervention, but existing protocols vary widely and often lack adaptation to sarcopenia severity. The present study aims to review the effectiveness of exercise protocols developed after the EWGSOP2 consensus and evaluate their adaptation to sarcopenia severity stages. Methods: This systematic review followed PRISMA guidelines. PubMed and Scopus were searched for studies published after the EWGSOP2 consensus involving participants of 65 years and over with primary sarcopenia and managed through exercise-only interventions. Risk of bias was assessed with the Cochrane Risk of Bias tool, and quality and transparency of exercise intervention were assessed with the Consensus on Exercise Reporting Template. Results: Ten studies met the inclusion criteria, with a total of 558 participants. Most interventions included resistance training, often within multicomponent programs. Statistically significant improvements were reported in muscle strength, mass, and physical performance. Additional benefits included enhancements in sleep quality, respiratory function, and specific biomarkers. However, only two studies classified sarcopenia severity, and reporting quality varied considerably. Conclusions: Exercise interventions, especially multicomponent and individualized protocols, are effective at improving outcomes related to sarcopenia in older adults. However, better alignment with diagnostic classifications and standardized reporting are needed to improve clinical translation and program replication.

## 1. Introduction

Aging is a physiological process that causes changes in people and is directly associated with progressive decline in body functions, including those related to muscle tissue [1]. The loss of muscle mass leads to negative consequences in older people such as falls, fractures, social isolation, functional decline, hospitalizations, and even mortality [1]. These physiological changes primarily involve alterations in body composition. With advancing age, muscle mass tends to decline, leading to changes in muscular strength and functional capacity. These aspects are closely related to the concept of sarcopenia. Sarcopenia is a progressive muscle disease associated with aging, characterized by a reduction in muscle strength and/or physical performance (e.g., gait speed), as well as a decrease in muscle mass [2].

With the increase in the number of older people, sarcopenia is expected to become an increasingly significant public health concern in the aging stage. Studies speak of a worldwide prevalence of sarcopenia of 10% in people over 60 years old and 50% in people over 80 years old [3]. However, these figures may vary depending on the diagnostic criteria used and the specific characteristics of the population studied. Due to this increase, several countries consider the importance of making changes in their health system. The approach to sarcopenia as a condition of aging must be preventive, effective, safe, and efficient [4].

In response to the increasing relevance of this condition, the European Working Group on Sarcopenia in Older People (EWGSOP) established diagnostic criteria in 2010 [2], later revised in the EWGSOP2 consensus to improve its identification and facilitate early diagnosis and management of sarcopenia in clinical practice [5].

The EWGSOP2 criteria use, as a pre-test, the Strength, Assistance in walking, Rise from chair, Climb stairs and Falls (SARC-F) questionnaire [6] to determine the risk of developing sarcopenia. To confirm the diagnosis and determine its severity, the EWGSOP2 algorithm is applied through a three-step process [2,7]. First, muscle strength is evaluated—typically using handgrip strength—as the primary indicator of possible sarcopenia (probable sarcopenia). Second, muscle quantity or quality is assessed to confirm the diagnosis (confirmed sarcopenia), provided that low muscle strength has already been identified. In older adults, the loss of appendicular skeletal muscle mass is commonly associated with sarcopenia and is often measured using Bioelectrical Impedance Analysis (BIA) or DEXA [8]. Third, physical performance is evaluated to determine the severity of sarcopenia (severe sarcopenia). This is typically measured using the Short Physical Performance Battery (SPPB) [9,10,11].

EWGSOP2 [2] recommends that therapeutic strategies for treating sarcopenia should primarily focus on restoring muscle strength, as its reduction is the central element of this disease [2]. Consequently, exercise interventions—particularly resistance training—are strongly encouraged for this population [11,12]. In 2017, Papa et al. [12] conducted a systematic review examining physical activity in older adults aimed at preventing sarcopenia. Their findings indicated that resistance training effectively helps preserve muscle mass, thereby reducing the risk of sarcopenia and falls in older individuals. Exercise also appears to improve outcomes in individuals already affected by sarcopenia: a more recent systematic review from 2021 concluded that combining resistance exercise with aerobic training represents the most effective strategy to improve muscle function [13]. Importantly, the benefits of exercise extend beyond muscular changes, contributing significantly to improved overall well-being. A recent meta-analysis [14] found that, in older adults with sarcopenia, there is high- to moderate-certainty evidence that resistance training—with or without nutritional support—and combined resistance, aerobic, and balance training are the most effective interventions for enhancing quality of life. These findings underline that physical exercise is not only a preventive and therapeutic strategy for preserving muscle mass and strength but also a key component in maintaining autonomy, psychological well-being, and social engagement in older adults. Importantly, all of these reviews consistently highlight that the effectiveness of such interventions depends heavily on their individualization [14].

Understanding key aspects of older adults’ health, such as sarcopenia, enables the development of more targeted interventions to mitigate associated risks. Scientific evidence suggests that reducing sedentary behavior and promoting structured, controlled physical exercise can help stabilize or even reduce the progression of sarcopenia and other age-related conditions [15,16]. However, to achieve these benefits, exercise programs must be carefully adapted in terms of intensity, volume, density, frequency, type of activity, and progression to align with the individual’s functional status, underlying health conditions, and personal preferences [17,18]. Ensuring both safety and adherence is essential. Tailoring exercise regimens to the specific needs and capacities of older adults is therefore critical to maximizing the effectiveness of interventions and promoting sustainable improvements in overall quality of life.

Therefore, to design an appropriate therapeutic physical exercise plan, it is first necessary to identify the degree of severity of sarcopenia according to the current EWGSOP2 criteria. From this classification, adapt the physical exercise to each category according to its morphological characteristics and define the parameters of the intervention. Despite the existing evidence, exercise protocols remain heterogeneous and often lack alignment with sarcopenia severity. It is still unclear which exercise modalities are most effective according to the severity of sarcopenia and whether published protocols are designed for individualized application. Existing systematic reviews do not take these criteria into account; therefore, it is necessary to review the current literature to synthesize the available evidence and provide a structured view on the effectiveness of different interventions.

The primary objective of this review is to evaluate the effectiveness of physical exercise interventions in improving sarcopenia-related outcomes in studies published since the establishment of the EWGSOP2 consensus. The secondary objective is to analyze whether the published protocols are tailored to be applied according to the severity of sarcopenia as established by the expert groups.

## 2. Materials and Methods

This review was carried out in adherence to the Preferred Reporting Items for Systematic Reviews and Meta-Analyses (PRISMA) guidelines [19]. As this study is based on synthesized data from original research, ethical approval was not deemed necessary. This systematic review is registered in the Open Science Framework registry (https://osf.io/phjdn, accessed on 29 May 2025).

### 2.1. Eligibility Criteria

The inclusion criteria for the search were as follows: (1) experimental studies (randomized controlled trials and quasi-experimental studies); (2) published in English; (3) released after the publication of the EWGSOP2 criteria (from 2019 onwards); (4) involving participants aged over 65 years; (5) diagnosing primary sarcopenia (attributed solely to aging); and (6) employing exercise as the sole intervention. Studies that did not meet these inclusion criteria were excluded.

### 2.2. Data Sources

The literature review was conducted between September 2024 and March 2025. An electronic search was performed using the PubMed and Scopus databases. A comprehensive search strategy was employed, integrating a combination of keywords and relevant subject headings to ensure sensitivity. The research question, formulated using the PICO framework (Participants, Intervention, Comparison, and Outcomes), was as follows: What is the effectiveness of exercise in managing sarcopenia among older adults, as assessed in studies published since the introduction of the EWGSOP2 criteria? Three groups of keywords were established, corresponding to: (1) “sarcopenia”; (2) “exercise”; and (3) “older adults”. Terms from the first category were combined with those from the second and third categories. The complete search strategy was as follows: (Sarcopenia OR “Muscle loss” OR “Muscle atrophy”) AND (“Exercise therapy” OR “Physical Activity” OR “Resistance Training” OR “Strength Training” OR “Aerobic Exercise”) AND (Older adults OR Elderly OR Aged). The literature search was systematically designed by two researchers (M.T.M. and E.M.M.) with specialized training in conducting systematic reviews.

### 2.3. Study Selection

All records identified through the search were imported into the Rayyan platform. First, duplicates were manually removed within the platform. Six researchers, organized in three pairs (C.P.M.-M.T.M., P.F.G.-A.E.S, and S.R.F.-E.M.M.) independently and blindly conducted the various phases of the selection process, beginning with the identification of relevant studies.

In the first phase, studies were screened based on their title and abstract to determine eligibility. In the second phase, the selected studies were examined in full text, alongside a review of their bibliographies. The authors evaluated each phase mentioned above based on the established inclusion and exclusion criteria. The result was a final set of full-text articles for data extraction and inclusion to analyze for risk of bias [20] and quality appraisal [21].

Any disagreements arising at any stage were resolved through joint discussion between the researchers. If consensus could not be reached, a third researcher was designated to review and resolve the conflict. However, in this review process, the intervention of a third researcher was not required.

### 2.4. Risk of Bias

The Cochrane Risk of Bias (RoB) tool was used to assess the risk of bias of the included randomized clinical trials [20]. This tool allows for a structured evaluation of multiple domains of potential bias, such as random sequence generation, allocation concealment, blinding, incomplete outcome data, selective reporting, and other sources of bias, thereby ensuring a comprehensive appraisal of the methodological quality of the studies. Discrepancies were resolved through discussion, and, when necessary, a third researcher was consulted to reach a final decision.

### 2.5. Quality Appraisal

To assess the quality appraisal, the Modified Consensus on Exercise Reporting Template (CERT) [21] was used as a checklist to ensure that all items related to the replication of physical exercise interventions were included in all the articles included in the review. The CERT, a 16-item checklist developed by an international panel of exercise experts, is designed to improve the reporting of exercise programs in all evaluative study designs and contains 7 categories: materials, provider, delivery, location, dosage, tailoring, and compliance.

### 2.6. Data Extraction

The authors worked independently and blindly, through a template, to extract the information from each study and prepare a descriptive table. The information included the following aspects: the authors and year of publication; the participants’ profiles, including the severity of sarcopenia; and the study groups and number of participants in each group. Additionally, detailed information regarding the intervention was collected, including the setting, whether the intervention was tailored to participants, the type of intervention, the duration of each session, the frequency of the sessions, and the total duration of the intervention. In addition, the results obtained in each study were extracted (Appendix A). This table presents each variable along with its measurement method, the pre- and post-intervention values for both the control and experimental groups, the follow-up assessments, and the significant differences observed both between and within groups.

## 3. Results

### 3.1. Overview of Studies

The flowchart in Figure 1 illustrates the search results and study selection process. After searching the available scientific literature across the databases, 463 studies were identified, of which 118 were excluded as duplicates. A total of 345 records were screened based on their titles and abstracts. Following this screening, 316 records were excluded. Consequently, 29 studies were assessed in full-text form, and 19 of these were excluded for not meeting the eligibility criteria. Ultimately, 10 studies that met all eligibility criteria were included in this systematic review.

### 3.2. Methodological Quality

Table 1 summarizes the risk of bias assessment across the included randomized controlled trials, evaluated using the RoB tool. Overall, 50% of the studies were rated as having a high risk of bias [22,23,24,25,26], while the remaining 50% raised some concerns [27,28,29,30,31]; notably, none of the studies achieved a low overall risk of bias. When analyzing individual domains, the highest methodological quality was observed in the domains of “Deviations from intended interventions” (90% low risk) and “Missing outcome data” (80% low risk). In contrast, the “Selection of the reported result” domain showed the most limitations, with 50% of studies rated low risk.

Table 2 presents the quality appraisal of exercise programs across all included evaluative study designs in exercise research. Overall, the quality of reporting was highly variable, with no study fully meeting all 16 CERT items. The studies by Flor-Rufino et al., Courel-Ibáñez et al. and Liang et al. [22,29,30] demonstrated the most comprehensive reporting, fulfilling 11, 10, and 9 criteria, respectively. In contrast, the studies by Sang-Jung et al., De Sá Souza et al., and Guo et al. [25,27,31] exhibited the poorest reporting quality, fulfilling 0, 1, and 3 items, respectively. Among the 16 CERT items, the most consistently reported domains were supervision of the exercise sessions (item 4) and the setting in which exercises were performed (item 12), which are described in half of the studies. However, critical components such as the use of motivational strategies (item 6), inclusion of illustrative materials for exercise replication (item 8), and intervention delivered and performed as planned (item 16) were largely underreported (in two studies). The item regarding the content of any home program component (item 9) received a low response rate because only two studies included a home program intervention, whereas the remaining studies did not. Notably, three studies provided information on whether the interventions were individualized (item 14), which is a key aspect for tailoring exercise to patient-specific needs.

### 3.3. Characteristics of the Studies

The characteristics of the included studies are summarized in Table 3, Appendix A. The articles had sample sizes ranging from 22 to 103, contributing to a total sample size of 558 participants, aged between 65 and 84 years. In 40% of the included studies, participants were exclusively women [22,23,25,28], while the remaining 60% included both men and women [24,26,27,29,30,31].

The settings in which the interventions were conducted included physical performance laboratories [22,23,25], participant’s homes [27,28], nursing homes [24,29], hospitals, and gyms. Of the studies included in the review, nine incorporated strength training: two as a standalone intervention [27,28], and seven as part of multicomponent exercise programs [22,23,25,26,29,30,31]. Other dimensions included in these programs were flexibility in eight studies [22,23,24,25,26,29,30,31], balance in six studies [24,25,26,29,30,31], and cardiorespiratory endurance [24,25,26,30,31], relaxation [22,23,24,26,31], and postural control [22,23,24,26,31] in five studies. In addition, education was used as a complementary tool in four studies [23,26,27,31].

The interventions varied in duration and frequency, with session lengths ranging from 25 to 75 min, a frequency of 2 to 5 days per week, and intervention periods lasting from a minimum of 12 weeks to a maximum of 31 weeks. Adherence levels in the included interventions varied between 83.3% and 100%. Further details in adherence levels and dropouts across intervention groups can be observed in Appendix A.

### 3.4. Outcome Measurements

The outcomes assessed included those related to muscle strength, body composition, physical performance, and other physiological or clinical parameters. Within the muscle strength domain, the most frequently used measures were handgrip strength, isometric knee extension strength, and dynamic lower limb strength exercises, such as leg press and knee extension. Upper and lower limb endurance tests were also commonly applied. In the body composition domain, the most commonly reported variables were total muscle mass, skeletal muscle index, appendicular skeletal muscle mass, and body composition parameters such as fat-free mass, body fat percentage, and BMI. For physical performance, the most utilized assessments included Gait speed, the SPPB and its subcomponents, the Timed Up and Go (TUG) test, the 30-Second chair stand, and the 6-Minute walk test. These tests were frequently used to evaluate functional mobility, balance, and overall physical function. Among other outcomes, the most frequently assessed included pulmonary function, quality of life, and sleep quality indicators. Additionally, a subset of studies incorporated biomarkers to explore underlying physiological mechanisms. Detailed information on outcome measurements can be found in Appendix A.

#### 3.4.1. Effectiveness of Physical Exercise Protocols

The interventions analyzed demonstrated notable improvements in muscle strength, body composition, and physical performance across several studies. Within the muscle strength domain, significant between-group gains were observed in handgrip strength, isometric and dynamic contractions, and muscular endurance in five of the trials [22,24,30,31]. All of these studies implemented multicomponent interventions that included resistance training, with the exception of the study by Myong-Won [28], which focused exclusively on resistance exercises. Between-group improvements in body composition were observed across four interventions [22,25,28,31], specifically in outcomes related to skeletal muscle mass, total muscle mass, fat mass, and waist-to-hip ratios. With the exception of Myong-Won’s study [28], which exclusively applied resistance training, all other studies combined resistance exercises with additional components as part of multicomponent interventions. In terms of physical performance, consistent improvements were found in tools like the Barthel Index [24,30] or in tests such as the 30-Second chair stand, 30-Second arm curl, Chair sit-and-reach, 8-Foot up-and-go, 2-Minute step test, Gait speed [28], SPPB [30], TUG, Berg Balance Scale, and 6-Minute walk test [26]. These gains were especially prominent in the studies by Ilke-Sen (2021), Myong-Won (2021), and Liang (2020) which implemented interventions including strength, balance, aerobic function, flexibility, and even relaxation and demonstrated both within- and between-group statistical significance [26,28,30].

Additionally, other outcomes were also registered. The intervention by de Sá Souza [27], which was multicomponent in nature, led to improvements in sleep-related outcomes and an increase in testosterone levels. Myong-Won’s intervention [28], focused exclusively on resistance training, resulted in significant improvements in the biomarker follistatin. Similarly, the multicomponent program implemented by Flor-Rufino [23] produced positive effects on respiratory function. The intervention by Hsiao-Ting [24], based on Tai Chi principles, enhanced joint range of motion across multiple areas. Additionally, Ilke-Sen’s [26] intervention led to improvements in quality of life. Notably, each of these studies was the only one among those reviewed to assess these specific outcomes.

The study by Courel-Ibáñez [29] did not show significant between-group differences; however, it reported within-group improvements in both strength and physical performance. Notably, both groups followed the Vivifrail multicomponent protocol but differed in the duration of the training and detraining phases. The summary of study results and significant within- and between-group differences can be found in Appendix A.

#### 3.4.2. Tailoring of Exercise Protocols to Individual Needs

The participants’ severity of sarcopenia is defined in 2 of 10 studies: Hsiao-Ting [24] indicates that the enrolled participants are diagnosed with the category of “probable sarcopenia”, following the latest Asian Working Group for Sarcopenia criteria [32]. Ilke-Sen [26] used the EWSOP2 criteria [5], including participants with “confirmed sarcopenia”.

Several of the included studies implemented individualized exercise interventions based on objective or functional criteria. Most notably, individualized load prescription was applied in six studies using percentages of one-repetition maximum (1RM), either through direct testing or submaximal estimation, with progressive adjustments throughout the intervention period. These included the protocols by Flor-Rufino, de Sá Souza, Liang, and Guo [22,27,30,31], where resistance training loads ranged from 40% to 85% of one-repetition maximum (1RM), and progression was based on periodic reassessment. Vivifrail, described by Courel-Ibáñez [29], applied a tailored multicomponent training model based on baseline functional capacity, assigning participants to one of four training levels. Other studies, such as those by Ilke-Sen [26] and Myong-Won [28], individualized the exercise intensity using subjective measures like the OMNI resistance scale and the Borg Rating of Perceived Exertion [32]. In contrast, the Vitality Acupunch program by Hsiao-Ting [24] followed a standardized, group-based format without individualized progression or load adjustments. Overall, most interventions integrated some form of personalization to optimize safety, adherence, and physiological outcomes in older adults. The complete details of the interventions can be found in Appendix A.

## 4. Discussion

This systematic review aimed to examine the effectiveness of physical exercise protocols in improving sarcopenia-related outcomes in older adults following the adoption of the EWGSOP2 criteria. A secondary objective was to assess whether these protocols were adapted based on the severity of sarcopenia. The findings confirm that structured physical exercise interventions, particularly those incorporating resistance training, are effective at improving muscle strength, body composition, and physical performance. Moreover, a substantial number of studies adopted individualized approaches, though only a minority explicitly defined participants’ sarcopenia severity based on EWGSOP2 or similar diagnostic frameworks.

Consistent with previous systematic reviews and meta-analyses [33,34], resistance training, either alone or as part of a multicomponent protocol, was found to be the most consistently effective modality in enhancing muscle strength. Notably, five studies demonstrated significant between-group improvements in strength-related outcomes, including handgrip strength, isometric and dynamic lower-limb strength, and muscular endurance. The study by Myong-Won [28], which implemented resistance training as a standalone intervention, produced meaningful gains in both strength and physical performance. In contrast, the remaining interventions combined resistance training with additional components such as balance, flexibility, or aerobic exercise. These multicomponent programs not only improved strength and performance but also had a positive impact on pulmonary function, range of motion, and muscle mass, underscoring their effectiveness in enhancing muscle quality. This finding aligns with recent global consensus on optimal exercise recommendations for enhancing healthy longevity in older adults (ICFSR) [35], which emphasize the superiority of combined modalities over isolated interventions in improving overall functional capacity in older adults. Properly structured and progressive training programs lasting 3 to 6 months have been shown to increase muscle strength by approximately 40% to 150%, depending on individual characteristics and training intensity [36]. In this review, the duration of effective interventions ranged from 3 to 8 months.

Following in terms of body composition, four interventions showed between-group improvements in key measures such as skeletal muscle index, total muscle mass, or fat composition. These changes were most commonly associated with protocols that included resistance training tailored to individual capacity using the OMNI resistance for active muscle scale and percentages of 1RM [22,25,28,31]. The consistent use of objective progression models, ranging from 40% to 85% of 1RM, demonstrates the value of load individualization, a principle strongly supported in the geriatric training literature [35]: A properly designed progressive resistance training can counteract age-related changes in contractile function, atrophy, and morphology of the aging human skeletal muscles [37]. Additionally, such programs can lead to gains of 1 to 3 kg in total lean body mass or increases of 10% to 30% in muscle fiber cross-sectional area [38]. Regarding decrease in fat mass, published literature [39] indicates that resistance and aerobic exercises generally result in significant reductions in fat mass; however, the combination of aerobic and resistance training has demonstrated superiority in reducing fat mass in older men compared to aerobic training alone. It happens in the included study of Sang-Jung [25].

Physical performance outcomes also improved across a wide range of functional tests, including the SPPB, 6-Minute walk test, TUG, and 30-Second chair stand. These improvements were particularly marked in four studies, which delivered interventions that combined strength training with components such as balance, aerobic capacity, flexibility, and even relaxation techniques. These findings underscore the added value of including different components in exercise programs for older adults with sarcopenia. Current evidence-based research indicates that this kind of program can effectively enhance strength, power, gait speed, functional ability, and skeletal muscle index in older adults with sarcopenia regardless of their living setting (whether in the community, hospitals, or long-term care facilities) [40], while no drug has yet translated clinically relevant improvements on physical performance [41]. 

Furthermore, a subset of studies reported significant gains in sleep quality, respiratory function, and biomarkers such as testosterone, follistatin, and inflammatory cytokines. However, as each of these outcomes was assessed in only a single study, these findings should be interpreted with caution due to their limited generalizability.

These findings may be associated with the high levels of adherence observed in the interventions included, which ranged from 83.3% to 100%.

An important issue observed in this review is the limited use of diagnostic stratification based on sarcopenia severity. Only 2 out of the 10 included studies explicitly defined participants’ sarcopenia status: Hsiao-Ting [24] used the Asian Working Group for Sarcopenia criteria to include individuals with probable sarcopenia, while Ilke-Sen [26] applied the EWGSOP2 criteria and enrolled participants with confirmed sarcopenia. In contrast, some observational studies [42,43,44] have applied the EWGSOP2 criteria; yet, these have not examined the alignment between diagnosis and intervention design within experimental settings. This gap reflects a disconnect between diagnostic rigor and intervention design despite the central role of severity-based classification in contemporary clinical recommendations. Given that one of the main aims of this review was to explore whether interventions are aligned with EWGSOP2 severity stages, this limited use of severity-based stratification must be highlighted as both a key finding and a significant limitation. This omission may hinder effective stratification in clinical settings, where different severities require tailored treatment intensities. To improve both research quality and clinical relevance, future studies should explicitly adopt EWGSOP2 severity staging and report how interventions are adjusted accordingly. Nevertheless, in the studies that did incorporate severity classifications, both interventions proved effective across all assessed outcomes, which may be attributed to the degree of individualization. In both cases, participants trained three times per week; however, the intervention for individuals with probable sarcopenia was longer in total duration but involved shorter sessions, while those with confirmed sarcopenia followed a shorter program with longer session lengths. This tailored approach may have contributed to the observed effectiveness in each group.

However, when examining the degree of intervention individualization, a more favorable picture emerges. Several studies demonstrated a commitment to tailoring exercise protocols based on objective or functional assessments: using 1RM testing, either directly or via submaximal estimations, to prescribe and progressively adjust training intensity; subjective intensity scales, such as to personalize training, and others, such as Courel-Ibáñez’s Vivifrail program [29], applied stratified training based on functional capacity levels. Established principles of exercise prescription, namely specificity, overload, and progression, should be used to deliver an effective exercise dose [14]. While the optimal training dose remains under debate, evidence [45] indicates that even low-frequency, moderate-intensity programs (1–2 sessions per week at 50% of 1RM) can improve outcomes in older adults with sarcopenia, although greater benefits may be achieved with higher intensities (70–85% of 1RM, 2–3 sessions per week). The interventions included in this review align with this evidence, featuring session lengths ranging from 25 to 75 min, training frequencies between 2 and 5 days per week, and durations spanning 12 to 31 weeks. Most protocols incorporated progression models, with training loads varying from 40% to 85% of 1RM.

Nonetheless, this review is not without limitations. First, the overall methodological quality of the included studies was moderate, with no study rated as low risk of bias across all domains. In fact, 50% of the studies were assessed as having a high overall risk of bias, while the other 50% raised some concerns; none achieved a low overall risk rating. Although certain domains, such as “Deviations from intended interventions” and “Missing outcome data,” showed relatively strong methodological performance, the “Selection of the reported result” domain presented the most notable limitations. Additionally, reporting quality, as assessed by the CERT checklist, was variable and often incomplete, particularly in aspects such as motivational strategies, exercise replication materials, and fidelity monitoring. The average CERT score across the included studies was 6 out of 16, reflecting suboptimal reporting quality. This inconsistency in reporting may limit the replicability and clinical applicability of the interventions. Moreover, a considerable number of interventions were conducted exclusively in female participants, and few studies performed sex-specific analyses. This gender imbalance restricts the generalizability of the findings, particularly to older men with sarcopenia, who may differ in their physiological responses, baseline functional status, and adherence patterns to exercise interventions. Future research should prioritize more balanced recruitment and explore potential sex-specific effects to improve the external validity of intervention outcomes. These omissions and imbalances may hinder both the reproducibility of the protocols and their relevance in diverse clinical settings. Finally, the heterogeneity of the interventions—regarding type, duration, intensity, and outcome measures—limits the possibility of making direct comparisons across studies.

Future research should aim to standardize reporting practices, clearly define participants’ sarcopenia severity using established criteria, and explore the long-term impact of tailored interventions. Aligning protocol design with diagnostic stratification and ensuring comprehensive program documentation will be crucial for advancing clinical translation and maximizing the benefits of exercise in aging populations. These findings suggest that while some studies adequately described core components of the training protocols, the overall reporting quality remains insufficient to ensure replicability and clinical translation. Greater adherence to CERT guidelines in future trials would enhance transparency, allow for better comparison across studies, and facilitate implementation in real-world settings.

## 5. Conclusions

This review confirms that physical exercise, particularly when multicomponent and individualized, is effective in improving muscle strength, mass, and physical performance in older adults with sarcopenia. Resistance training was the most consistently effective component. However, few studies classified sarcopenia severity, which limits alignment between diagnosis and intervention design. Standardizing reporting and tailoring protocols based on diagnosis are key steps for advancing clinical application.

## Figures and Tables

**Figure 1 geriatrics-10-00091-f001:**
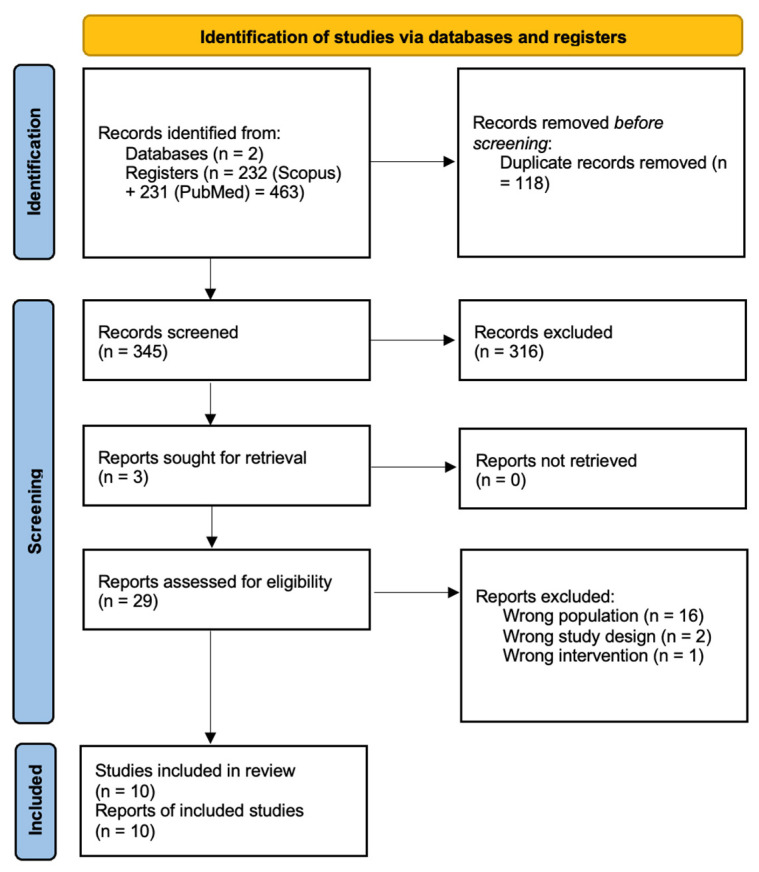
PRISMA flowchart.

**Table 1 geriatrics-10-00091-t001:** Cochrane Collaboration’s tool for assessing risk of bias.

Authors (Year)	Randomization Process	Deviations from Intended Interventions	Missing Outcome Data	Measurement of the Outcome	Selection of the Reported Result	Overall Bias
Courel-Ibáñez, J. (2022) [29]	+	+	+	?	+	?
De Sá Souza, H. (2022) [27]	?	?	+	+	?	?
Flor-Rufino, C. (2023a) [22]	+	+	-	+	?	-
Flor-Rufino, C. (2023b) [23]	+	+	-	+	?	-
Guo, H. (2024) [31]	?	+	+	+	+	?
Hsiao-Ting, T (2021) [24]	+	+	+	?	-	-
Ilke-Sen, E. (2021) [26]	?	+	+	-	+	-
Liang, Y. (2020) [30]	?	+	+	+	+	?
Myong-Won, S. (2021) [28]	+	+	+	?	+	?
Sang-Jung, W. (2019) [25]	+	+	+	-	+	-

“+”: Low Risk; “-“: High Risk; “?”: Some concerns.

**Table 2 geriatrics-10-00091-t002:** Quality appraisal of the included articles with the Modified Consensus on Exercise Reporting Template (CERT).

Authors (Year)	Items from the Modified CERT Template															
	1	2	3	4	5	6	7	8	9	10	11	12	13	14	15	16
Courel-Ibáñez, J. (2022) [29]	No	Yes	Yes	Yes	Yes	No	Yes	Yes	No	Yes	No	Yes	No	Yes	Yes	Yes
De Sá Souza, H. (2022) [27]	Yes	No	No	No	No	No	No	No	No	No	No	No	Yes	No	Yes	No
Flor-Rufino, C. (2023a) [22]	Yes	Yes	No	Yes	Yes	No	Yes	No	No	Yes	Yes	Yes	No	No	No	No
Flor-Rufino, C. (2023b) [23]	No	Yes	No	Yes	Yes	No	Yes	No	No	Yes	Yes	Yes	Yes	No	No	Yes
Guo, H. (2024) [31]	No	No	No	No	No	No	No	No	No	No	No	No	Yes	No	No	No
Hsiao-Ting, T. (2021) [24]	No	No	No	Yes	Yes	No	No	Yes	No	No	No	No	No	Yes	No	No
Ilke-Sen, E. (2021) [26]	No	No	Yes	No	Yes	Yes	No	No	Yes	Yes	Yes	Yes	No	No	No	No
Myong-Won, S. (2021) [28]	Yes	Yes	No	Yes	No	No	No	No	Yes	No	No	Yes	Yes	Yes	No	No
Liang, Y. (2020) [30]	No	Yes	Yes	Yes	No	Yes	No	No	No	Yes	Yes	Yes	Yes	Yes	Yes	No
Sang-Jung, W. (2019) [25]	No	No	No	No	No	No	No	No	No	No	No	No	No	No	No	No

Item 1—Provide equipment manufacturer, city, state, country, if appropriate, and appropriate copyright; Item 2—If exercise program is administered by multiple therapists, provide detail on how each therapist was trained in the intervention; Item 3—Whether exercises are performed individually or in a group; Item 4—Whether exercises are supervised or unsupervised; Item 5—Measurement and reporting of adherence to exercise; Item 6—Details of motivation strategies; Item 7—Decision rules for progressing the exercise program; Item 8—Each exercise is described so that it can be replicated (egg, illustrations, photographs); Item 9—Content of any home program component; Item 10—Non-exercise components; Item 11—How adverse events that occur during exercise are documented and managed; Item 12—Setting in which exercises are performed; Item 13—Detailed description of the exercises (egg, sets, repetitions, duration, intensity); Item 14—Whether exercises are generic (“one size fits all”) or tailored to the individual; Item 15—Decision rules that determines the starting level for exercise; Item 16—Whether the exercise intervention is delivered and performed as planned.

**Table 3 geriatrics-10-00091-t003:** Characteristics of each of the studies.

Authors (Year)	Participant Profiles/Severity of Sarcopenia *	Number of Participants and Groups	Details of the Intervention					
			Setting	Tailored Intervention	Type	Session Duration	Frequency	Duration of Intervention
Courel-Ibáñez, J. (2022) [29]	Men and women between 84–87 years/NA	*n* = 24 (2 groups) IG = 12 CG = 12	Nursing Home	Yes	IG: Multicomponent intervention CG: Multicomponent intervention	30–60 min	5 days/week	30 weeks
de Sá Souza, H. (2022) [27]	Men and women 65 years/NA	*n* = 28 (2 groups) IG = 14 CG = 14	Home	Yes	IG: Resistance intervention CG: Education	Based on sets and repetitions	3 days/ week	12 weeks
Flor-Rufino, C. (2023a) [22]	Women 70 years/NA	*n* = 51 (2 groups) IG = 24 CG = 27	Physical performance laboratory	Yes	IG: Multicomponent intervention CG: Phone follow-ups	65 min	IG = 2 days/week recovery time = 72 h	30 weeks
Flor-Rufino, C. (2023b) [23]	Women 70 years/NA	*n* = 51 (2 groups) IG = 24 CG = 27	Physical performance laboratory	Yes	IG: Multicomponent intervention CG: No intervention. Encouraged to stay active	65 min	IG = 2 days/week 72 h recovery period	30 weeks
Guo, H. (2024) [31]	Men and women between 65–75 years/NA	*n* = 93 (2 groups) IG = 63 CG = 30	Hospital	Yes	IG: Multicomponent intervention CG: Education	30 min	3 days /week	24 weeks
Hsiao-Ting, T. (2021) [24]	Men and women 65 years/probable sarcopenia	*n* = 103 (2 groups) IG = 52 CG = 51	Nursing home	No	IG: Multicomponent intervention CG: Routine daily activities	40 min	3 days/week	31 weeks
Ilke-Sen, E. (2021) [26]	Men and women between 65–80 years/confirmed sarcopenia	*n* = 100 (2 groups) IG = 50 CG = 50	Home	No	IG: Multicomponent intervention CG: Education + usual physical activity lifestyle	60 min	3 days /week	12 weeks
Liang, Y. (2020) [30]	Men and women between 80–90 years/NA	*n* = 60 (2 groups) IG = 30 CG = 30	Hospital (post-acute care unit)	Yes	IG: Multicomponent intervention CG: Education + resistance exercise program	50 min	2 days /week	12 weeks
Myong-Won, S. (2021) [28]	Women 65 years/NA	*n* = 22 (2 groups) IG = 12 CG = 10	Gym	Yes	IG: Resistance intervention CG: No intervention	60 min	3 days/ week	16 weeks
Sang Jung, W. (2019) [25]	Women between 75–80 years/NA	*n* = 26 (2 groups) IG = 13 CG = 13	Physical performance laboratory	Yes	IG: Multicomponent intervention CG: Usual physical activity lifestyle	25–75 min	IG: 3 days /week CG: 2 days /week	12 weeks

IG: intervention group, CG: control group, NA: not applicable. * According to the latest European Working Group on Sarcopenia in Older People/Asian Working Group for Sarcopenia categories.

## Data Availability

Data are contained within the article and the Appendix A.

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
