# Peer review of "Deepening Physical Exercise Intervention Protocols for Older People with Sarcopenia Following Establishment of the EWGSOP2 Consensus: A Systematic Review"

_geriatrics, 2025, doi:10.3390/geriatrics10040091_

Round 1

Reviewer 1 Report

Comments and Suggestions for Authors

Overall, the paper is well-written, organized, and contributes to knowledge regarding the efficacy of exercise interventions in persons with sarcopenia. The authors uncovered an important gap in the current state of knowledge, namely that the effectiveness of exercise interventions is not routinely matched to sarcopenia severity.

The primary objective of this systematic review was to provide an overview of the effectiveness of exercise protocols in improving sarcopenia since the establishment of the EWGSOP2 consensus. The secondary objective was to analyze whether the published protocols are tailored to be applied to the severity of sarcopenia.  The rationale for the review was that although exercise interventions are known to improve muscle mass, strength, and physical function in persons with sarcopenia, the effectiveness cannot be determined without knowing the severity of the sarcopenia at study entry and the criteria by which sarcopenia was evaluated. Individualization of exercise is expected to improve responses to the interventions, but individualization is not reported consistently. Furthermore, use of CERT to assess the quality of the exercise interventions in research has been used sporadically and could improve understanding the details needed to evaluate intervention effectiveness. As a result, it is not clear that exercise interventions are matched to the severity of sarcopenia.

The methods were clear; PRISMA guidelines were followed and the PRISMA checklist was completed accurately. Selection criteria for the relevant publications was explained and the search results presented in Figure 1.

The results were presented clearly in table and text formats, including risk of bias and CERT appraisal. The key characteristics of the included publications were included in the main manuscript with additional details provided in supplementary tables 1 and 2. The outcomes assessed were related to muscle mass and strength, physical performance, and other physiological or clinical parameters unique to certain studies. The effectiveness of the exercise interventions was described overall and for each study or intervention type (e.g., resistance training only or combined with aerobic exercise). Severity of sarcopenia was defined in 2 of the 10 studies, and tailoring of exercise protocols was addressed. The cited literature includes some “legacy” papers in the introduction section (more than 5 years since publication) and more recent papers. The studies included in the analyses were published after 2019, when the EWGSOP2 criteria were published.

The Discussion was presented logically and clearly, results were synthesized and interpreted in the context of sarcopenia, exercise, and aging. The key message was the gap that exists between the rigor of sarcopenia assessment (level of severity) and the exercise intervention design. This is an important finding because the EWGSOP2 consensus and CERT are readily available to assess each side of the gap. The findings could be influential for future studies and the interpretation of published studies. The limitations of the study were described, and the conclusion follows logically from the results.

Specific Comments:

Just a few about the supplementary data.

Table S1

Courel-Ibanez, J (2022). It is not clear whether the significance of the group differences (IG and CG) is from preintervention to postintervention or to end of follow-up.

Flor-Rufino, C (2023)a, Myong-Won S (2021), and Sang-Jung (2019): the muscle mass category would be more appropriately titled “body composition” because other variables such as fat mass and waist circumference are also reported.

Myong-Won S (2021) and Sang-Jung (2019): “free fat mass” change to “fat-free mass”

Author Response

Comments 1: Courel-Ibanez, J (2022). It is not clear whether the significance of the group differences (IG and CG) is from preintervention to postintervention or to end of follow-up.

Response 1: Thank you for this valuable observation. We agree that the results presented in Table S1 were potentially unclear. We have now revised the table to improve its interpretability. To better reflect the impact of the interventions, we have included only the pre- and post-intervention results.

Comments 2: Flor-Rufino, C (2023)a, Myong-Won S (2021), and Sang-Jung (2019): the muscle mass category would be more appropriately titled “body composition” because other variables such as fat mass and waist circumference are also reported.

Response 2: Thank you for this helpful suggestion. We agree that “body composition” is a more accurate term given the inclusion of variables such as fat mass and waist circumference. This change has now been incorporated both in the main text and in the corresponding table.

Comments 3: Myong-Won S (2021) and Sang-Jung (2019): “free fat mass” change to “fat-free mass”

Response 3: Thank you for pointing this out. The term has been corrected from “free fat mass” to the more accurate “fat-free mass” throughout the files.

Reviewer 2 Report

Comments and Suggestions for Authors

The manuscript addresses a timely and clinically relevant question regarding the alignment of exercise interventions with sarcopenia severity post-EWGSOP2 consensus. The adherence to PRISMA guidelines and inclusion of risk of bias (RoB) and CERT assessments are commendable. However, several issues related to clarity, methodological rigor, reporting, and depth of analysis limit the manuscript’s current impact and reproducibility.

Major concerns:

1. Lack of Clarity in Research Question Framing

Issue: The primary and secondary aims are not sharply delineated.

Location: Page 2, Lines 119–123

Suggestion: Clearly state whether this review is assessing effectiveness, protocol reporting quality, or alignment with sarcopenia severity—or all three. The current statement is too broad and blends aims.

2. Sarcopenia Severity Under-addressed Despite Being a Focus

Issue: Only 2 of 10 studies classify sarcopenia severity, yet the review's premise is to evaluate exercise protocols in light of EWGSOP2's severity classification.

Location: Page 11, Lines 322–326

Suggestion: This should be emphasized as a major finding and limitation in both the abstract and conclusion. Further, consider excluding studies not using EWGSOP2 severity stages or separate their analysis.

3. Inconsistent Reporting of CERT and RoB Scores

Issue: The text discusses CERT and RoB results without consistent interpretation across studies.

Location: Pages 7–8, Tables 1 & 2

Suggestion: Include aggregate statistics (mean CERT score, proportion of studies meeting ≥10 criteria) and visualizations for transparency.

4. Poor Standardization of Terminology

Issue: The manuscript inconsistently uses terms like "confirmed sarcopenia," “probable,” and “primary sarcopenia” without consistent definitions.

Location: Pages 3–5, 10–11

Suggestion: Add a glossary or define each diagnostic level (EWGSOP2) in the Introduction and apply terms uniformly.

Minor Concerns:

A. Abstract Language and Grammar

Issue: “be er alignment” instead of “better alignment”

Location: Page 1, Line 39

Suggestion: Carefully proofread for typographical and formatting errors throughout.

B. Figure 1 Formatting

Issue: The PRISMA figure is cited but not included in readable format.

Location: Page 6, Line 207

Suggestion: Include a properly formatted, high-resolution PRISMA flow diagram and ensure all figure legends are complete.

C. Missing Discussion on Adherence and Safety

Issue: Little mention of adherence rates, adverse events, or dropouts.

Location: Discussion section, Page 12

Suggestion: These are essential for clinical translation. Include a table summarizing adherence and dropout rates.

D. Gender Bias in Studies

Issue: Several studies focus exclusively on women.

Location: Page 9, Lines 252–255

Suggestion: Discuss this as a limitation affecting generalizability, particularly to older men with sarcopenia.

Author Response

Comments 1:  Lack of Clarity in Research Question Framing

Issue: The primary and secondary aims are not sharply delineated.

Location: Page 2, Lines 119–123

Suggestion: Clearly state whether this review is assessing effectiveness, protocol reporting quality, or alignment with sarcopenia severity—or all three. The current statement is too broad and blends aims.

Response 1: Thank you for this helpful comment. The primary and secondary objectives have now been clearly defined in the manuscript.

The primary objective is to evaluate the effectiveness of physical exercise protocols used in studies published since the release of the EWGSOP2 consensus in improving sarcopenia-related outcomes. The secondary objective is to examine whether these protocols are adapted or aligned with the severity stages of sarcopenia as defined by experts.

The assessment of methodological quality is not an objective of the review itself, but rather a requirement of the PRISMA guidelines. Therefore, it is included in the Methods section and reported in the Results to support interpretation, but it does not represent a central aim of the study.

Comments 2: Sarcopenia Severity Under-addressed Despite Being a Focus

Issue: Only 2 of 10 studies classify sarcopenia severity, yet the review's premise is to evaluate exercise protocols in light of EWGSOP2's severity classification.

Location: Page 11, Lines 322–326

Suggestion: This should be emphasized as a major finding and limitation in both the abstract and conclusion. Further, consider excluding studies not using EWGSOP2 severity stages or separate their analysis.

Response 2: Thank you for this valuable observation. The limited use of EWGSOP2 severity classification across the included studies has been emphasised as a key finding in both the Discussion and Conclusions sections. The studies that did incorporate sarcopenia severity classification are now discussed in a dedicated paragraph within the Discussion, where we highlight not only their methodological distinction but also the effectiveness of their interventions across all measured outcomes. This focused analysis aims to illustrate the potential benefits of tailoring interventions according to severity, and further supports the recommendation for adopting EWGSOP2 staging in future research.

Comments 3. Inconsistent Reporting of CERT and RoB Scores

Issue: The text discusses CERT and RoB results without consistent interpretation across studies.

Location: Pages 7–8, Tables 1 & 2

Suggestion: Include aggregate statistics (mean CERT score, proportion of studies meeting ≥10 criteria) and visualizations for transparency.

Response 3: Thank you for this observation. This information was already detailed in the Results section; however, it has now been further expanded and explicitly discussed in the Limitations section, particularly with regard to the distribution of risk of bias across studies and the variability in reporting quality as assessed by the CERT checklist.

Comments 4. Poor Standardization of Terminology

Issue: The manuscript inconsistently uses terms like "confirmed sarcopenia," “probable,” and “primary sarcopenia” without consistent definitions.

Location: Pages 3–5, 10–11

Suggestion: Add a glossary or define each diagnostic level (EWGSOP2) in the Introduction and apply terms uniformly.

Response 4: Thank you for this important observation. We have now added the EWGSOP2 diagnostic terminology in the Introduction to clarify each diagnostic level. Additionally, we have defined the criteria for primary sarcopenia within the eligibility criteria to ensure consistent use of terminology throughout the manuscript.

Comments 5. A. Abstract Language and Grammar

Issue: “be er alignment” instead of “better alignment”

Location: Page 1, Line 39

Suggestion: Carefully proofread for typographical and formatting errors throughout.

Response 5: Thank you for highlighting this typographical error. We have corrected “be er” to “better” in the Abstract, and have carefully proofread the entire manuscript to address any similar issues.

Comments 6. B. Figure 1 Formatting

Issue: The PRISMA figure is cited but not included in readable format.

Location: Page 6, Line 207

Suggestion: Include a properly formatted, high-resolution PRISMA flow diagram and ensure all figure legends are complete.

Response 6. Thank you for this helpful observation. We have used the official PRISMA template to generate a properly formatted and high-resolution flow diagram, which is now included in the main manuscript. Additionally, as a precautionary measure, the figure has also been provided as a separate supplementary file to ensure it can be easily accessed and reviewed.

Comments 7. C. Missing Discussion on Adherence and Safety

Issue: Little mention of adherence rates, adverse events, or dropouts.

Location: Discussion section, Page 12

Suggestion: These are essential for clinical translation. Include a table summarizing adherence and dropout rates.

Response 7. Thank you for your suggestion. The table summarising adherence and dropout rates has now been included as Supplementary Table 2. In addition, relevant data have been integrated into both the Results and Discussion sections to support interpretation and contextualise the findings.

Comments 8. D. Gender Bias in Studies

Issue: Several studies focus exclusively on women.

Location: Page 9, Lines 252–255

Suggestion: Discuss this as a limitation affecting generalizability, particularly to older men with sarcopenia.

Response 8. Thank you for this thoughtful suggestion. We have now expanded the discussion of this point in the Limitations section. Specifically, we acknowledge that the inclusion of studies focusing exclusively on women may restrict the generalisability of the findings, particularly to older men with sarcopenia, who may respond differently to exercise interventions. This limitation has been elaborated to highlight the need for more sex-balanced research in future studies.

Round 2

Reviewer 2 Report

Comments and Suggestions for Authors

Thank you for addressing my concerns.